# Antioxidant Effect of Wheat Germ Extracts and Their Antilipidemic Effect in Palmitic Acid-Induced Steatosis in HepG2 and 3T3-L1 Cells

**DOI:** 10.3390/foods10051061

**Published:** 2021-05-12

**Authors:** Humna Liaqat, Kyeong Jin Kim, Soo-yeon Park, Sung Keun Jung, Sung Hee Park, Seokwon Lim, Ji Yeon Kim

**Affiliations:** 1Department of Food Science and Technology, Seoul National University of Science and Technology, 232, Gongneung-ro, Nowon-gu, Seoul 01811, Korea; humnaliaqat710@gmail.com (H.L.); sunghpark@seoultech.ac.kr (S.H.P.); 2Department of Nano Bio Engineering, Seoul National University of Science and Technology, 232, Gongneung-ro, Nowon-gu, Seoul 01811, Korea; jinnykim@seoultech.ac.kr; 3Lab of Nanobio, Seoul National University of Science and Technology, 232, Gongneung-ro, Nowon-gu, Seoul 01811, Korea; sooyeon.park@seoultech.ac.kr; 4School of Food Science and Biotechnology, Kyungpook National University, Daegu 41566, Korea; skjung04@knu.ac.kr; 5Department of Food Science and Biotechnology, College of Bionano Technology, Gachon University, 1342, Seongnam-daero, Sujung-gu, Seongnam-si 13120, Korea; slim@gachon.ac.kr

**Keywords:** wheat germ extracts, bioactive compounds, antioxidants, antilipidemic effect

## Abstract

Wheat germ (WG) is a by-product of wheat milling and comprises many bioactive compounds. This study aimed to compare the antioxidant and antilipidemic effects of different WG extracts (WGEs) by analyzing candidate bioactive compounds such as carotenoids, tocopherols, γ-oryzanol, and biogenic amines by reversed-phase high-performance liquid chromatography. Antioxidant activity was determined using the ABTS, DPPH, and FRAP assays. The antilipidemic effect was evaluated in palmitic acid-induced steatosis in HepG2 hepatocytes and 3T3-L1 adipocytes. Cellular lipid accumulation was assessed by Oil Red O staining and a cellular triglyceride content assay. All analyzed WGEs showed significant antioxidant potential, although some bioactive compounds, such as carotenoids, tocopherols, and γ-oryzanol, were the highest in the ethanol extract. Correlation analysis revealed the antioxidant potential of all identified biogenic amines except for spermidine. Ethanol and n-hexane extracts significantly inhibited cellular lipid accumulation in cell models. These results suggest that WGEs exhibit promising antioxidant potential, with a variety of bioactive compounds. Collectively, the findings of this study suggest that bioactive compounds in WGEs attenuate plasma lipid and oxidation levels. In conclusion, WG can be used as a natural antioxidant and nutraceutical using appropriate solvents and extraction methods.

## 1. Introduction

Many physiological systems of the human body are damaged by reactive oxygen species (ROS). These ROS are generated due to several internal (mitochondrial respiratory chain, NADPH oxidase, and xanthine oxidase) and external (stress, smoke, air pollutants, radiation, etc.) factors [1]. Excessive ROS levels lead to oxidative stress, which causes various cellular dysfunctions. Moreover, oxidative stress has been shown association with many pathological conditions related to metabolic syndromes, such as hypertension, hyperglycemia, hyperlipidemia, and cancer [2]. Non-alcoholic fatty liver disease (NAFLD) is an emerging chronic liver disease, which interferes with lipid metabolism, resulting in increased lipid synthesis and excessive fat accumulation in the liver (hepatic steatosis), leading to a wide variety of hepatic dysfunctions [3]. Moreover, NAFLD is closely associated with obesity [4,5]. Obesity is a well-known metabolic disease caused by the excessive accumulation of fat in adipocytes, and adipogenesis, the process of differentiation of the preadipocytes to mature adipocytes, is involved in intracellular lipid accumulation [6]. Therefore, the approaches of reducing obesity and liver lipid accumulation have attracted interest for limiting NAFLD development [7]. Clinical treatment for NAFLD using hepatoprotective agents, such as antihypertensive, cell-protective, and antihyperlipidemic agents as well as antioxidants, has shown promising results [8]. Furthermore, reducing oxidative stress and inhibiting adipogenesis have also been suggested to help treat and prevent obesity [9,10].

In recent decades, numerous clinical trials and epidemiological studies have established an association between metabolic diseases and unhealthy lifestyles [11,12]. Concomitantly, several studies have also reported the potential of dietary supplementation of functional foods in mitigating metabolic disorders [13]. In this context, the extracts of whole wheat (*Triticum aestivum*) obtained using different solvents, such as water, methanol, and ethanol, have been explored to evaluate its antioxidative activities and potential health benefits [14,15,16]. Wheat, an important whole grain commodity, is one of the most consumed foods worldwide. Whole grain cereals are distinguished from refined cereals by the presence of bran and germ fractions, both of which are rich in bioactive compounds.

Wheat germ (WG) is typically discarded during the milling process, but it has a high concentration of nutrients [17]. It contains approximately 52% carbohydrates, 23% protein, 11% water, and 10% lipids. The bioactive compounds in WG mainly include antioxidants, such as flavonoids, polyphenols, tocopherols, tocotrienol (Vitamin E; known for its health benefits), carotenoids, and plant sterols, and biogenic amines, especially polyamines [18]. Biogenic amines in food matrices have attracted growing interest regarding the nutritional paradox and are well recognized for their antioxidant capacity [19]. Diets enriched in these compounds provide many benefits for human health (promotion of fat metabolism and regulation of blood pressure and neurotransmitters), but excess consumption may cause food poisoning. Some biogenic amines, such as putrescine, spermine, spermidine, histamine, and cadaverine, are included as industrial food quality parameters in the Chemical Quality Index [19]. Polyamines such as putrescine, spermine, and spermidine maintain healthy metabolic function [20]. The antioxidant capacity of polyamines mainly affects membrane lipids and nucleic acids and can eliminate free radicals, especially in lipophilic media [18]. WG has one of the highest polyamine contents among foodstuffs [21].

Additionally, γ-oryzanol, another phenolic compound, is a group of ferulic acid esters of triterpene alcohols and phytosterols. γ-oryzanol exhibits strong antioxidant potential and lowers lipid absorption, resulting in an improved lipid profile [22]. The therapeutic effects of γ-oryzanol have been explored in several health issues in humans, including cancer, hyperlipidemia, diabetes, and inflammation-related diseases [23].

Several studies have reported the beneficial effects of WG on various health aspects; however, only a few studies have reported its effects on metabolic markers [8]. Therefore, this is first study, which evaluated the polyphenol and flavonoid contents of various WG extracts (WGEs) approved by the food industry, using organic solvents and water extracts. Additionally, we quantified candidate bioactive compounds in different WGEs, determined the antioxidant effects, and assessed the antiadipogenic effects using two cell lines.

## 2. Materials and Methods

### 2.1. Chemicals and Reagents

We obtained 2,4,6-tris(2-pyridyl)-1,3,5-triazine (TPTZ) from Showa Chemical Co., Ltd. (Tokyo, Japan). The Folin–Ciocalteu phenol reagent, catechin, ascorbic acid, 2,2-diphenyl-1-picrylhydrazyl (DPPH), 2,2-azino-bis(3-ethylbenzothiazodine-6-sulfonic acid) (ABTS), dansyl chloride, bovine serum albumin (BSA), and palmitic acid (PA) were purchased from Sigma-Aldrich (St. Louis, MO, USA). Standards of biogenic amines, carotenoids, tocopherols, and γ-oryzanol were also obtained from Sigma-Aldrich. All HPLC-grade solvents were purchased from J.T. Baker, Inc. (Deventer, The Netherlands). Dulbecco’s modified Eagle’s medium (DMEM), fetal bovine serum (FBS), Dulbecco’s phosphate-buffered saline (DPBS), and penicillin-streptomycin mixture were procured from Biowest (Nuaille, Cholet, France). A buffer solution of 4-(2-hydroxyethyl)-1-piperazineethanesulfonic acid (HEPES) (1 M) was obtained from Gibco (Rockville, MD, USA).

### 2.2. Preparation of Wheat Germ Extracts

Raw WG was provided by Daehan Flour Mills Co., Ltd. (Incheon, Korea) and stored at −20 °C. Six different WGEs were prepared using different extraction methods: thermal, subcritical, and ultrasound-assisted water extractions and solvent extractions using acetone (50% *v*/*v*), n-hexane (50% *v*/*v;* hexane), and ethanol (70% *v*/*v*) [24,25]. The composition of the WG is shown in Table 1. In brief, WG was homogenized with 20× the volume of each solvent, including distilled water (DW). For the solvent extractions, homogenized solutions were stirred for 24 h at room temperature (RT). Thermal extraction to produce the hot water extract (HWE) was performed by heating a homogenized DW solution at 200 °C for 20 h. The subcritical water extract (SWE) was produced using a customized pilot-scale subcritical processing reactor equipped with an internal electrical heater (1.5 kW) and outer cooling jacket. SWE was prepared by heating the extract for a 30 min holding time (total processing time: 150 min) at 200 °C at a pressure of 20 MPa. For the ultrasonic water extract (UWE), an ultrasonic processor (Sonics & Materials, Inc., Newtown Square, PA, USA) was used at 80% amplitude at 500 W for 6 h. The extracts were then centrifuged at 3000× *g* at 4 °C for 10 min. The supernatants were concentrated using a rotary evaporator (UT-1000; EYELA, Tokyo, Japan), and the obtained extracts were lyophilized for 96 h and stored at −20 °C. After the complete preparation of samples, the yield of each WGEs were calculated as mg extract/g WG. The yield of WGEs is: acetone, 20.07; hexane, 24.76; ethanol, 21.94; HWE, 18.08; SWE, 23.63; UWE, 25.56 (mg extract/g WG). For the antioxidant capacities and some bioactive compounds (total polyphenols and flavonoids) analysis, WGEs were dissolved in DW, sonicated, filtered, and then clear solutions were stored at 4 °C until further analysis.

### 2.3. Quantification of Total Polyphenolic Content

Total phenolic content was determined using the Folin–Ciocalteu assay [1]. Gallic acid was used as the standard, and Folin–Ciocalteu reagent (160 μL) was added to the sample solution and incubated for 5 min. Then, 300 μL of sodium carbonate solution (10%) was added, and the solution was allowed to stand at RT for 30 min. Subsequently, the absorbance was measured at 750 nm. The total phenol content is expressed as milligrams of gallic acid equivalent per gram of the extract.

### 2.4. Quantification of Total Flavonoid Content

The aluminum chloride colorimetric method was used to determine the total flavonoid content using catechin as a standard [26]. One hundred microliters of sample solution were mixed with 30 μL of NaNO_2_ (5%), and the mixture was diluted with 400 μL of water. After 5 min, 10% AlCl_3_ (30 μL) was added, and the mixture was allowed to stand for 6 min at RT. Then, 1 M NaOH (200 μL) diluted with water (240 μL) was added, and the absorbance was measured at 510 nm. The total flavonoid content is expressed as milligrams of catechin equivalent per gram of the sample.

### 2.5. Quantification of Bioactive Compounds by Reversed-Phase High-Performance Liquid Chromatography (RP-HPLC)

#### 2.5.1. Preparation of Standard Stock and Working Solution

The chemical standards (external standards) of biogenic amines included tryptamine, putrescine dihydrochloride, cadaverine dihydrochloride, histamine dihydrochloride, tyramine hydrochloride, spermine tetrahydrochloride, spermidine trihydrochloride, and 1,7-diaminoheptane. Lutein, zeaxanthin, β-cryptoxanthin, α-carotene, and β-carotene were used as external carotenoid standards, and α-tocopherol and γ-tocopherol were the external standards for tocopherols. Stock solutions for all standards were prepared at a concentration of 10,000 mg/L, and the running solutions (1000 mg/L) at different concentrations (0–50 mg/L) were prepared in methanol. A solution of 1,7-diaminoheptane (1 mg/mL) was used as an internal standard (IS) for the detection of biogenic amines. All of the above standard solutions were kept frozen at −20 °C until further analyses.

#### 2.5.2. Quantification of Biogenic Amines

Pretreatment of Samples: Biogenic amine extraction was carried out according to the method described by Ben-Gigirey et al. [27]. WGE (100 mg) was added to 2 mL of 0.4 M perchloric acid (PCA) at 4 °C for 2 h and centrifuged at 3000× *g* for 5 min to obtain a clear supernatant. The residue was extracted and combined with the former supernatant.

Derivatization of Samples and Standards: Derivatization of biogenic amines was performed according to a previously described method with minor modifications [28]. The IS (20 μL) was spiked in 1 mL of sample extract (prepared in the above sample pretreatment) and standard solution. A solution of 200 μL of NaOH (2 M) and 300 μL of saturated NaHCO₃ was added to this mixture. Then, 2 mL of a dansyl chloride solution (10 mg/mL) prepared in acetone was added and incubated at 40 °C for 45 min (in dim light because derivatives of dansyl chloride are light sensitive). Subsequently, 100 μL of 25% NH₄OH was added before incubation for 30 min at RT. The volume of the incubated solution was adjusted to 5 mL with acetonitrile. Finally, the solution was centrifuged at 3000× *g* for 5 min, and the supernatant was filtered using 0.45 μm filters (Woongki Science Co., Ltd., Seoul, Korea). The filtered sample was stored at −25 °C for HPLC analysis.

Chromatography Conditions: HPLC analyses were performed using a Shiseido SI-2 series HPLC system (Shiseido, Tokyo, Japan). Separation of biogenic amines was achieved using Shiseido Capcell Pak C18 MG II S-5 (5 µm, 4.6 × 250 mm; Waters, Milford, MA, USA). The column effluent was measured at 254 nm. The mobile phase comprising 0.1 M ammonium acetate (Solvent A) and acetonitrile (Solvent B) was used at a flow rate of 1 mL/min with a sample injection of 20 μL. A linear gradient elution starting from 50:50 (A: B) at 0 min to 10:90 at 19 min, followed by isocratic elution at 50:50 from 20 to 35 min was used for detection of the peaks [29].

#### 2.5.3. Quantification of Carotenoids, Tocopherols, and γ-Oryzanol

Extraction by Saponification: Carotenoids, tocopherols, and γ-oryzanol were extracted according to a previously reported method [30,31] with some modifications. To the sample extracts (200 mg), 5 mL of ethanolic pyrogallol (60 g/L) as an antioxidant, 2 mL of ethanol (95%), 2 mL of NaCl (10 g/L), and 2 mL of KOH (600 g/L) were added. The prepared solutions were then placed in a water bath at 70 °C for 45 min and mixed every 10 min for saponification. Then, the solution was diluted with 15 mL of NaCl (10 g/L). The suspension was extracted with 5 mL of hexane/ethyl acetate (9:1 *v*/*v*), vortexed, and centrifuged at 3000× *g* for 10 min. Re-extraction was performed twice or thrice until the supernatant was clear. The organic supernatant was dried under a stream of nitrogen. The dried residue was diluted with methanol and then filtered. Carotenoids are light sensitive; therefore, they were tested under dim light.

Chromatography Conditions: Chromatographic separation of carotenoids was achieved using an RP-HPLC system equipped with an analytical YMC C18 carotenoid column (5 µm, 4.6 × 250 mm; Waters, Milford, MA, USA). Spectrophotometric detection was achieved using a diode array detector set in the range of 350–500 nm. Peaks were detected at 450 nm. The column was eluted with a binary gradient using the following HPLC solvents: (A) methanol/methyl tert-butyl ether (MTBE) (95:5, *v*/*v*) and (B) methanol/MTBE/water (8:90:2, *v*/*v*/*v*) containing 2.6 mM/L ammonium acetate. The following gradient profile was used for the separation: 0–5 min, 5% B; 8–17 min, 10–45% B; 20–22 min, 57.2–30% B; 25–30 min, 30–50% B. The flow rate was 1 mL/min, and the injection volume was 20 μL.

Tocopherols and γ-oryzanol were separated by RP-HPLC equipped C18 (5 µm, 4.6 × 250 mm). The HPLC mobile phase consisted of three solvents: (A) methanol, (B) water, and (C) butanol, and the separation was achieved following an isocratic elution program: 92% A, 4% B, and 4% C with a flow rate of 1 mL/min and injection volume of 20 μL [32]. Tocopherols were detected at 292 nm, and γ-oryzanol was detected at 325 nm. Chromatograms were recorded, and peak areas were used to calculate the content of tocopherols and γ-oryzanol compared with those of the standards.

### 2.6. Linearity and Lower Limit of Detection (LOD)

The linear range is the range of concentrations in which the HPLC signals are directly proportional to the concentration of the analyte in the sample. The linear range was determined using the dilution of 10 standards from 0.01 to 10 mg/L. For more accuracy, the experiments were performed over four consecutive days. The calibration curve was plotted against all concentrations of the standards. The LOD was the lowest concentration that could be differentiated from the background noise (3 is the approximate signal-to-noise ratio).

### 2.7. Total Antioxidant Capacity of WGEs

#### 2.7.1. ABTS Assay

The ABTS assay was performed according to a previously described method [1]. A solution of 5 mL of ABTS (7 mM) and 88 μL of potassium persulfate (140 mM) was prepared. This solution (ABTS reagent) was incubated for 16 h in the dark at RT. The stock solution was diluted with ethanol to obtain an absorbance at 734 nm. Then, 50 μL of each sample (10 mg/mL) was added to 1 mL of ABTS reagent. After 3 min, the absorbance was measured at 734 nm. The ABTS value was calculated as follows:

ABTS radical-scavenging activity (%) = [1 − (A_1_/A_0_)] × 100, where A_0_ is the absorbance of the blank, and A_1_ is the absorbance of the sample.

#### 2.7.2. DPPH Assay

The DPPH assay was performed according to a previously described method [1,33,34]. To evaluate the DPPH radical-scavenging activity, the DPPH was diluted to 0.1 mM with ethanol. Then, 50 μL of each sample (10 mg/mL) was added to DPPH reagent (150 μL). After 30 min in the dark, the absorbance was measured at λ = 517 nm using a spectrophotometer (BioTek Instruments, Inc., Winooski, VT, USA). The DPPH radical-scavenging activity was calculated as follows:

DPPH radical-scavenging activity (%) = [1 − (A_1_/A_0_)] × 100, where A_0_ is the absorbance of the blank and A_1_ is the sample absorbance.

#### 2.7.3. Ferric Reducing Antioxidant Power (FRAP) Assay

The FRAP assay is another method for detecting antioxidant capacity. FRAP reagent was prepared using a previously described method [1,35]. In brief, 300 mM acetate buffer (pH 3.6), 10 mM TPTZ (2,4,6-tripyridyl-s-triazine) solution in 40 mM hydrochloric acid, and 20 mM iron (III) chloride were mixed in a 10:1:1 ratio. FRAP reagent (150 μL) was mixed with 20 μL of the sample, 20 μL of ascorbic acid (positive control), and 20 μL of DW (blank). The absorbance of each was measured at 593 nm. The FRAP value was calculated using the following equation:

FRAP value = [(A_1_ − A_0_)/(A_c_ − A_0_)] × 2, where A_c_ is the absorbance of the positive control, A_1_ is the absorbance of the sample, and A_0_ is the absorbance of the blank.

### 2.8. Assessment of Antilipidemic Effects of WGEs In Vitro

#### 2.8.1. Cell Culture

HepG2 (human hepatoma cell line) and 3T3-L1 (murine preadipocytes of fibroblast cell line) were purchased from the Korea Cell Line Bank (KCLB, Seoul, Korea). HepG2 cells were used to examine the inhibitory effect of WGEs on hepatic lipid accumulation. HepG2 cells were cultured in DMEM supplemented with 2% penicillin-streptomycin, 2% HEPES, and 10% FBS. 3T3-L1 cells were used to determine the effects of WGEs on differentiated adipocytes. The 3T3-L1 cells were cultured in media containing DMEM, 10% bovine serum (BS), and 1% penicillin-streptomycin. Both cells were maintained incubated at 37 °C in a 5% CO_2_ atmosphere. Media renewal in both cell lines was performed every 2–3 days, and sub-culturing was performed until 80% confluence.

#### 2.8.2. PA-Induced Steatosis in HepG2 cells

The PA solution was prepared by dissolving 100 mM PA (stock solution) in DW at 70 °C. The PA solution was filtered and stored at 4 °C until use. Then, PA solution (5 mM) was added to serum-free DMEM containing 5% fatty acid-free BSA. For the induction of steatosis, HepG2 cells were seeded in 12-well plates at a density of 1 × 10⁵ cells/mL with the culture media. After reaching 80–90% confluence, cells were treated with serum-free media containing PA solution (0.5 mM) and samples (100 μg/mL) for 24 h.

#### 2.8.3. Differentiation of 3T3-L1 Cells

Pre-adipocyte 3T3-L1 cells (1 × 10^5^ cells/well) were seeded in 6-well plates and treated with BS media until they reached 100% confluence. To induce differentiation in preadipocytes, they were exposed to samples (100 μg/mL) and cultured in a differentiated medium containing the following: DMEM, 10% FBS, 1% penicillin-streptomycin, 1 μM dexamethasone, 10 μg/mL insulin, and 115 μg/mL 3-isobutyl-1-methylxanthine (IBMX). After six days of culture, the cells were transferred from differentiated media to insulin media (DMEM supplemented with 10% FBS, 1% penicillin-streptomycin, and 10 μg/mL insulin) and maintained for another six days.

### 2.9. Cellular Lipid Accumulation by Oil Red O (ORO) Staining

ORO staining was performed to examine the morphology and total lipid accumulation in the HepG2 and 3T3-L1 cells. To prepare the ORO stock solution, ORO was dissolved in 0.5% isopropanol and diluted with isopropanol and water (6:4). Cells were washed twice with PBS and fixed with 4% paraformaldehyde at RT (HepG2 for 30 min and 3T3-L1 for 60 min). The fixed cells were rinsed with PBS and stained with ORO for 30 min at RT. After washing, images of the stained cells were captured using an inverted microscope (Nikon, Tokyo, Japan). Then, the dyed lipids were quantified by dissolving them in isopropanol. Leaching liquor (200 μL) was injected into 96-well culture plates, and the absorbance was measured at 510 nm.

### 2.10. Cellular Triglyceride Content

Cellular triglyceride (TG) content was quantified according to a previously described method with some modifications [36,37]. In brief, HepG2 and 3T3-L1 cells were seeded in 12 plates (1 × 10^5^ cells/well). After treatment, the cells were washed with iced DPBS and scraped using a rubber policeman. Then the cells were collected by centrifugation at 210× *g* for 10 min. Cellular lipids were extracted using a chloroform/isopropanol/Tween 20 (7:11:0.1) solution by vortexing. The solution was centrifuged at 13,000× *g* for 10 min, and the supernatant was collected. The organic solvent was removed under nitrogen, and the dried lipids were diluted with isopropanol in 10% Triton X-100. The cellular TG was analyzed using Asan kits (according to the manufacturer’s instructions), whereas cellular protein content was measured using the Bradford protein assay. The results were expressed as milligrams of triglyceride per gram of cellular protein.

### 2.11. Statistical Analysis

Pearson’s correlation coefficient (r) was evaluated using the R software package. Results are expressed as the average value ± standard deviation (X ± S) for triplicate determinations. The significance level was determined by one-way ANOVA using Duncan’s multiple range test (SPSS 20; SPSS Inc., Chicago, IL, USA). Differences were considered statistically significant at *p* < 0.05.

## 3. Results

### 3.1. Total Polyphenol and Flavonoid Contents

The total flavonoid contents in all samples extracted with different extraction methods were highly proportional to the total polyphenolic contents, meaning that when there was a higher content of total polyphenols, there was also a higher content of total flavonoids. Of the six extracts, the total polyphenol (128.62 mg) and flavonoid (21.29 mg) contents were the highest in the SWE and the lowest in the UWE (67.60 and 9.38 mg, respectively) (Table 2). These findings indicate that the antioxidant potential of SWE, with higher total polyphenol and flavonoid contents, could be higher than that of the other WGEs.

### 3.2. Analysis of Biogenic Amines

The chemical structures of the identified biogenic amines (external standards) are shown in Figure 1. Several methods have been published to quantify biogenic amines from fermented food products, especially plant-based foods [38]. This study identified seven biogenic amines (tryptamine, putrescine, cadaverine, histamine, tyramine, spermidine, and spermine) from six WGEs. Biogenic amines were identified by their retention times compared with standard solutions. The retention times of the standards were stable and consistently reproducible. Representative chromatograms of the standard solutions and one of the samples are presented in Figure 2.

The quantified biogenic amines are shown in Table 3. The content of putrescine and histamine was significantly higher in the hexane extract (194.16 and 50.16 mg/kg) than in the other extracts. The contents of cadaverine (65.34 mg/kg) and tyramine (38.48 mg/kg) were markedly higher in the acetone and ethanol extracts, respectively, than in the remaining extracts. The HWE showed the highest content of spermidine and spermine among all samples, with values of 234.91 and 62.87 mg/kg, respectively. The UWE sample showed the lowest quantities of putrescine, cadaverine, spermidine, and spermine, whereas its tryptamine content (229.35 mg/kg) was the highest among the extracts; histamine and tyramine were not detected in the UWE. The R^2^ and LOD values of the validated HPLC method for biogenic amines are shown in Table 4.

### 3.3. Analysis of Carotenoids, Tocopherols, and γ-Oryzanol

The contents of carotenoids, tocopherols, and γ-oryzanol in the WGEs are shown in Table 5. These bioactive compounds were identified by their retention time compared with the standard solutions. The retention time of the standards was stable and reproducible. The representative standard chromatograms of all the analyzed bioactive compounds and representative chromatograms of one of the observed samples are presented in Figure 3A–F.

As shown in Table 5, the profiles of bioactive compounds were not uniform in all extracts. For example, lutein and zeaxanthin were detected in four of the six WGEs. Lutein and zeaxanthin were the main carotenoids present, whereas the carotenoids β-cryptoxanthin, α-carotene, and β-carotene were not detected.

The main tocopherol in the WG samples was α-tocopherol. The highest content of α-tocopherol was in the ethanol extract (312.17 mg/kg), whereas it was the lowest in the hexane extract (5.92 mg/kg). α-Tocopherol was not detected in the other extracts. Similarly, γ-tocopherol and γ-oryzanol were detected only in three WGEs (ethanol, hexane, and UWE) at ranges of 22.78–228.37 and 486.49–5928.23 mg/kg, respectively. The γ-tocopherol and γ-oryzanol contents in the ethanol extract were much higher than those in the other samples were. No carotenoids, tocopherols, or γ-oryzanol were detected in the HWE or SWE samples. The R^2^ and LOD values of the validated HPLC method for these bioactive compounds are shown in Table 6.

### 3.4. Antioxidant Activities

To measure the antioxidant activity of WGEs, we analyzed their radical-scavenging activity and reducing power. ABTS, DPPH, and FRAP assays are widely used to detect the antioxidant capacities of natural products [39]. Total 10 mg/mL amount of WGEs are used in all antioxidant activities as shown in Table 7. SWE showed the highest ABTS radical-scavenging activity (87.54%), and UWE had the lowest value among all the samples (40.49%). The DPPH activity measures the electron-donating ability and ranged from 10.54% to 66.97%. The FRAP value determines the reduction of Fe^3+^ to Fe^2+^ ions by donor electrons in the sample. The highest FRAP value was obtained with the hexane extract (0.95%), whereas it was the lowest for the UWE (0.20%).

### 3.5. Cellular Lipid Accumulation and TG Content

The in vitro studies were carried out using hepatocyte and adipocyte models in HepG2 and 3T3-L1 cell lines [9]. In this study, PA was used to induce hepatic steatosis, and cells were treated with WGEs for 24 h to evaluate lipid accumulation by ORO staining.

The morphology of stained cells with lipid droplets is shown in Figure 4A. The results demonstrated the potential of PA to induce lipid accumulation. In the control group, ORO staining showed no steatosis; however, after treatment with PA, lipid droplets accumulated in the cytoplasm of HepG2 cells. Compared with the PA-induced group, the organic solvent extractions of WG, especially the ethanol and hexane extracts, reduced lipid droplet accumulation and significantly attenuated lipid fusion in the HepG2 cells (Figure 4B).

To examine the antiadipogenic effect of WGEs, 3T3-L1 preadipocytes were treated with the six WGEs for 12 days. The results of ORO staining for 3T3-L1 cell morphology are shown in Figure 5A. No lipid droplets were observed in the control group (preadipocytes). In addition, lipid accumulation was significantly reduced in the ethanol and hexane extract groups compared with that in the differentiated adipocyte group (MDI; differentiated media; positive control) (Figure 5B). In summary, among all the WGEs, the ethanol and hexane extracts effectively and significantly alleviated lipid accumulation in both cell lines. However, there was no significant effect of the water WGEs in either cell line. 

In adipocytes, lipid droplets have a core of lipid esters, and these lipid ester cores contain TGs. To evaluate the inhibition of TG accumulation, we measured cellular TG levels directly in HepG2 and 3T3-L1 cells. The content of TG was significantly reduced in ethanol and hexane extracts (treatment groups) in both cell types, similar to the quantification of total lipid reduction by ORO staining, as shown in Figure 6. Other water extracts did not show any significant effect in reducing the cellular TG content.

### 3.6. Correlation of Antioxidant Capacity and Biogenic Amines

Some studies have suggested the association between antioxidant activities and biogenic amines in various food commodities [40]. Therefore, in this study, we assessed the correlation between the antioxidant and antilipidemic capacities of the bioactive compounds identified in the WGEs to verify their relationships. Figure 7 presents the r-value (Pearson’s correlation coefficient) between 1.0 and −1.0, which shows the strength or weakness of correlation between the analyzed factors. Red (1.0), blue (−1.0), and white (0.0) colors show positive, negative, and no correlation, respectively.

Fourteen bioactive compounds were used for the analysis to establish the correlation between the compounds in the WGEs and their antioxidant and antilipidemic capacities. The antioxidant activities (ABTS, DPPH, and FRAP) of the individual bioactive compounds showed a similar trend of relationship (either positive, negative, or independent); however, the strength of correlation varied. Most of the bioactive compounds (12) were positively correlated, whereas tryptamine showed a strong negative correlation, and spermidine content was not associated with antioxidant capacity.

Antilipidemic factors were expressed in the form of lipid accumulation in hepatocytes and adipocytes (LA_hepatocyte and LA_adipocyte). It was observed that LA_hepatocyte and LA_adipocyte factors were negatively correlated with bioactive components. For example, lutein, zeaxanthin, α-tocopherol, γ-tocopherol, and γ-oryzanol were negatively correlated with lipid accumulation, resulting in lipid content reduction by some WGEs. These results indicated that with an increase in bioactive compounds, the LA_hepatocyte and LA_adipocyte factors decreased, increasing the antilipidemic activity and vice versa. These results suggest the potential of the bioactive compounds identified here contributing to the antioxidant and antilipidemic effects of WGEs.

These correlation findings indicate that the content of flavonoids, tyramine, histamine, and cadaverine are strongly positively correlated with antioxidant activities. Therefore, these bioactive compounds might be responsible for the antioxidant capacity of WG. In addition, some bioactive compounds, such as lutein, zeaxanthin, α-tocopherol, γ-tocopherol, and γ-oryzanol, are negatively correlated with lipid accumulation, which resulted in some WGEs showing reduced lipids. These bioactive compounds were mostly detected in the ethanol and hexane extracts as compared with the other extracts. These bioactive compounds might be responsible for the reduced cellular lipid accumulation in WGEs.

## 4. Discussion

In the present study, we employed six different kinds of extraction for the preparation of WG samples. Usually, conventional high-efficiency extraction methods employ toxic solvents including hexane, which requires a high degree of purification and is harmful to the environment. Moreover, a limited number of solvents are allowed for food and drug applications [41]. However, HWE, SWE, and UWE have been proposed as promising efficient techniques with several benefits, such as high purity, economic feasibility, and lack of toxicity, compared with other extraction methods [41,42,43]. These methods are natural treatments that do not use dangerous chemicals and can be used widely as thermal treatments.

We evaluated the antioxidant and antilipidemic effects of six types of WGEs and analyzed their bioactive compound profiles, including polyphenols, flavonoids, carotenoids, tocopherols, and biogenic amines. Several studies in recent years used these bioactive compounds to reduce oxidative stress and plasma lipid levels. For example, biogenic amines have been investigated, especially for their antioxidant capacity [44,45]. A study reported that the antioxidant and antilipidemic effects correlated with the levels of phenolic compounds and tocopherols in mice [46]. Another study suggested that plasma lipid levels (triglycerides and cholesterol) are associated with circulating carotenoids [47,48], tocopherols, and tocotrienol [49]. Furthermore, randomized controlled trials reported that γ-oryzanol reduced cholesterol levels and increased antioxidant capacity in humans [50] and rats [51] with hyperlipidemia.

In this study, WGEs exhibited potent antioxidant activity, attributed to their total phenolic, flavonoid, biogenic amine, carotenoid (lutein and zeaxanthin), α-tocopherol, γ-tocopherol, and γ-oryzanol contents. Several studies have demonstrated that polyphenolic and flavonoid contents are related to antioxidant activity [1,52]. Accordingly, studies have also shown that lutein and zeaxanthin are the most abundant carotenoids in wheat [53]. Moreover, the carotenoids and tocopherols were mainly present in the organic solvents (acetone, ethanol, and hexane), possibly because these compounds are fat-soluble.

Excessive production of ROS is associated with oxidative stress and increased lipid accumulation during adipocyte differentiation [54]. Moreover, excess oxidative stress coincides with fat accumulation in the adipocytes. Therefore, reduction of excessive ROS levels and inhibition of adipocyte differentiation are important preventive strategies to alleviate oxidative stress. Thus, oxidative stress is an intriguing concept for the development of novel therapeutics, and we believe that the WGEs with potential antioxidant activities could be useful in this aspect.

Steatosis is the accumulation of visible lipid droplets in more than 5% of hepatocytes [55]. PA induces fat accumulation and inflammation [56]; therefore, it is used to induce steatosis in HepG2 cells. Under some physiological conditions, more than 95% of fat from ingested food is accumulated in adipose tissue and stored in the form of triglycerides [57]. Therefore, accelerating TG decomposition has been explored as a potential strategy to reduce lipid accumulation in hepatocytes. The significant decrease in lipid and TG levels exerted by the ethanol and hexane extracts suggests the potential antilipidemic effect of WGEs.

Nevertheless, the different WGEs contained varying contents of the bioactive compounds, but the antioxidant capacity of the ethanol extract was constantly significant in all assays. In addition, the water extracts (HWE, UWE, and SWE) did not have significant antihyperlipidemic effects; in contrast, the organic extracts, especially those of ethanol and hexane, significantly reduced lipid levels. The correlation analysis also demonstrated that the antioxidant and antilipidemic effects of WGEs might be attributed to their potent bioactive components. Some bioactive compounds were strongly correlated positively with the antioxidant effects and negatively with the antihyperlipidemic effects of WGEs.

The association between phytochemicals and disease prevention has been a major focus of health researchers for almost half a century [58,59,60]. Several studies have demonstrated that naturally occurring bioactive compounds and phytochemicals have a broad spectrum of physiological effects, including alleviation of metabolic disorder, inflammation, and oxidative stress [61,62,63]. These studies suggest that the medicinal effects of various food commodities could provide new ways to develop dietary strategies to prevent many diseases. However, the business of phytochemicals and their protective effect will only flourish successfully if clinical research can incorporate credible science with consumer demand, convenience, and awareness about the preventive role of dietary products in the development of non-communicable diseases.

To the best of our knowledge, this is the first study to quantify many phytochemicals, including carotenoids, tocopherol, polyphenols, flavonoids, biogenic amines, and γ-oryzanol. We included and compared all possible extraction methods and solvents to prepare WGEs because previous studies have stated that WG may not affect hyperlipidemia [64,65]. Nevertheless, this study has limitations; many biogenic amines present in WGEs correlated with antioxidant capacity, but we did not elucidate the relationships between other phytochemicals and the antioxidant capacity. We also did not determine the bioactive substances in WGEs responsible for the antilipidemic effect or the direct correlation between antioxidant and antilipidemic activities.

## 5. Conclusions

In conclusion, this study demonstrated that various WGEs containing bioactive compounds have high antioxidant capacity. Among the WGEs, the ethanol and hexane extracts significantly limited excessive fat accumulation in HepG2 and 3T3-L1 cells. Additionally, the correlation summary revealed that the bioactive compounds present in the WGEs might be responsible for their antioxidant and antilipidemic effects. Our results provide evidence that WG could serve as a source of natural antioxidants and nutraceuticals by using appropriate solvent and extraction processes. However, precise identification of the key bioactive compounds responsible for the antioxidant and antilipidemic effects of WGEs, as well as their underlying mechanism, is warranted.

## Figures and Tables

**Figure 1 foods-10-01061-f001:**
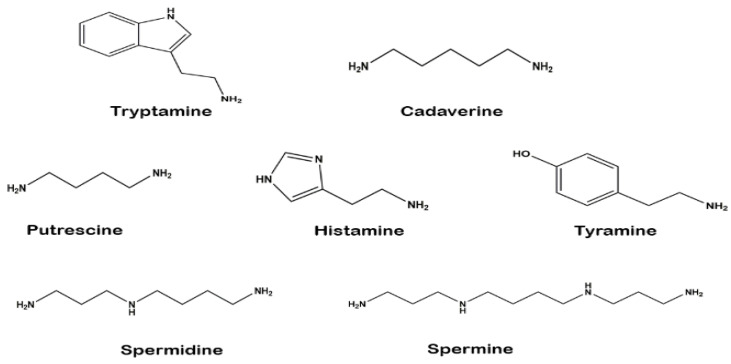
Chemical structures of seven identified biogenic amines.

**Figure 2 foods-10-01061-f002:**
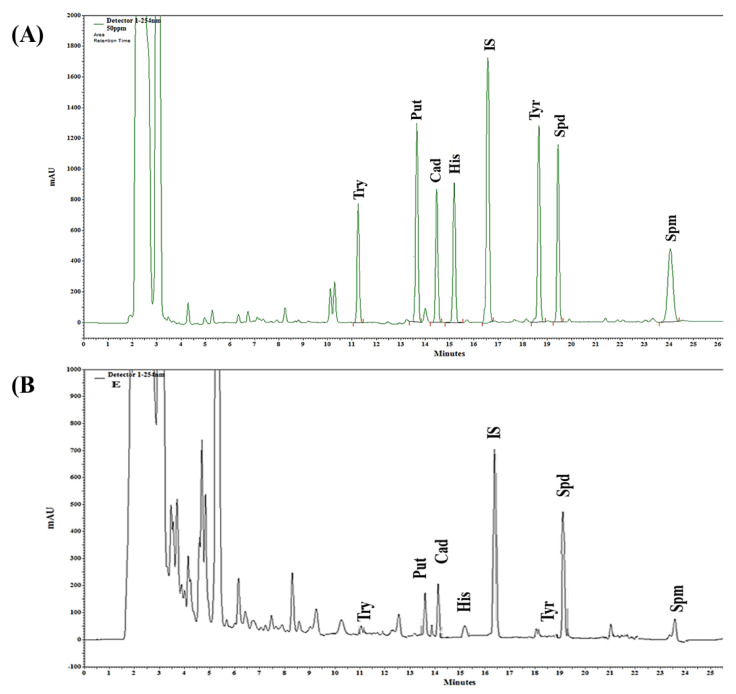
Representative HPLC chromatograms of all analyzed biogenic amines at λ = 254 nm. (**A**) Chromatogram of the standard mixture. (**B**) Representative chromatogram of one of the samples (ethanol extract). Try, tryptamine; Put, putrescine; Cad, cadaverine; His, histamine; IS (internal standard); Tyr, tyramine; Spd, spermidine; Spm, spermine.

**Figure 3 foods-10-01061-f003:**
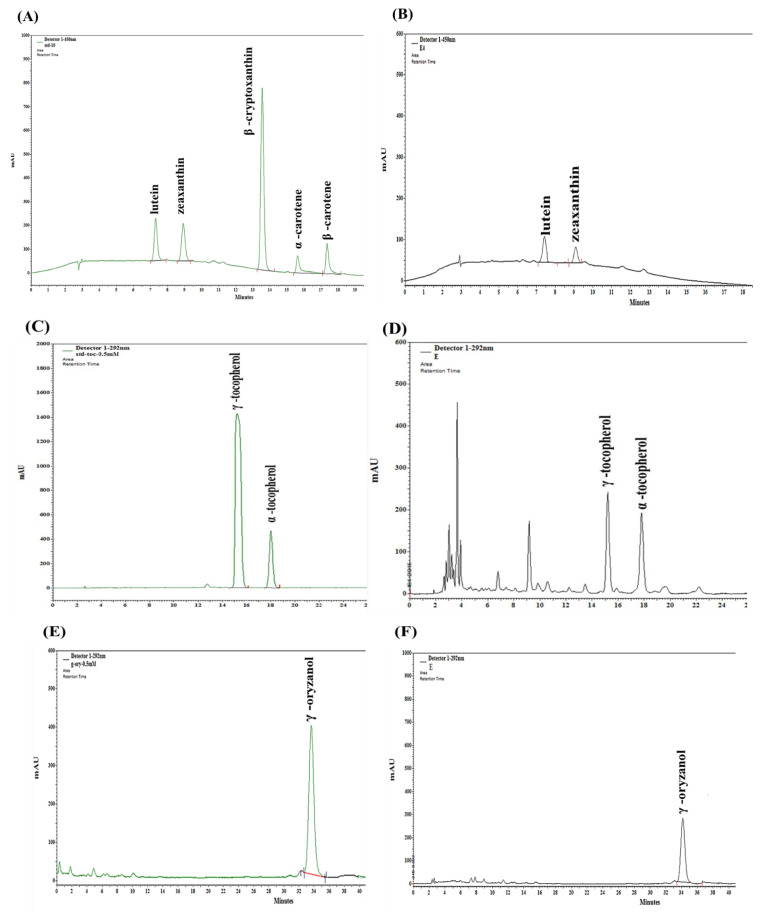
Representative HPLC chromatograms of bioactive compounds. (**A**) A standard mixture of all the analyzed carotenoids at λ = 450 nm. (**C**) A standard mixture of tocopherols at λ = 292 nm. (**E**) A standard mixture of γ-oryzanol at λ = 325 nm. (**B**,**D**,**F**) are representative chromatograms (ethanol extract) of carotenoids, tocopherols, and γ-oryzanol, respectively.

**Figure 4 foods-10-01061-f004:**
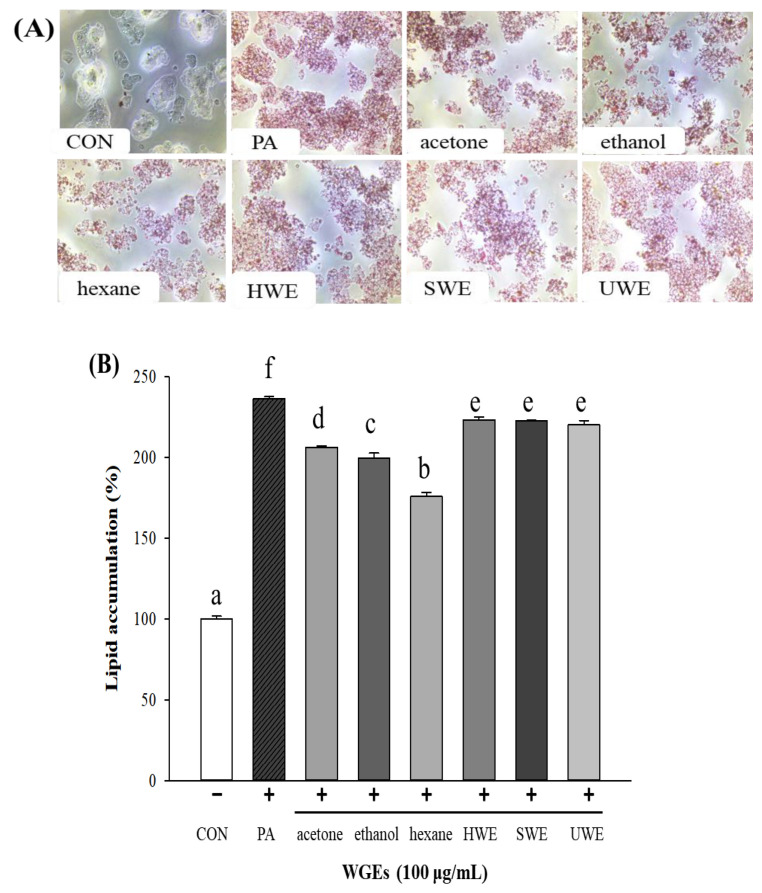
Effect of wheat germ extracts (WGEs) on cellular lipid accumulation in palmitic acid (PA)-induced steatosis in HepG2 hepatocytes. (**A**) The morphology of stained intracellular lipid droplets by ORO staining at 400× magnification. (**B**) The content of cellular lipid accumulation stained by Oil Red O (ORO). Different letters indicate significant differences at *p* < 0.05 determined by Duncan’s multiple range test. CON, control; HWE, hot water extract; SWE, subcritical water extract; UWE, ultrasonic water extract.

**Figure 5 foods-10-01061-f005:**
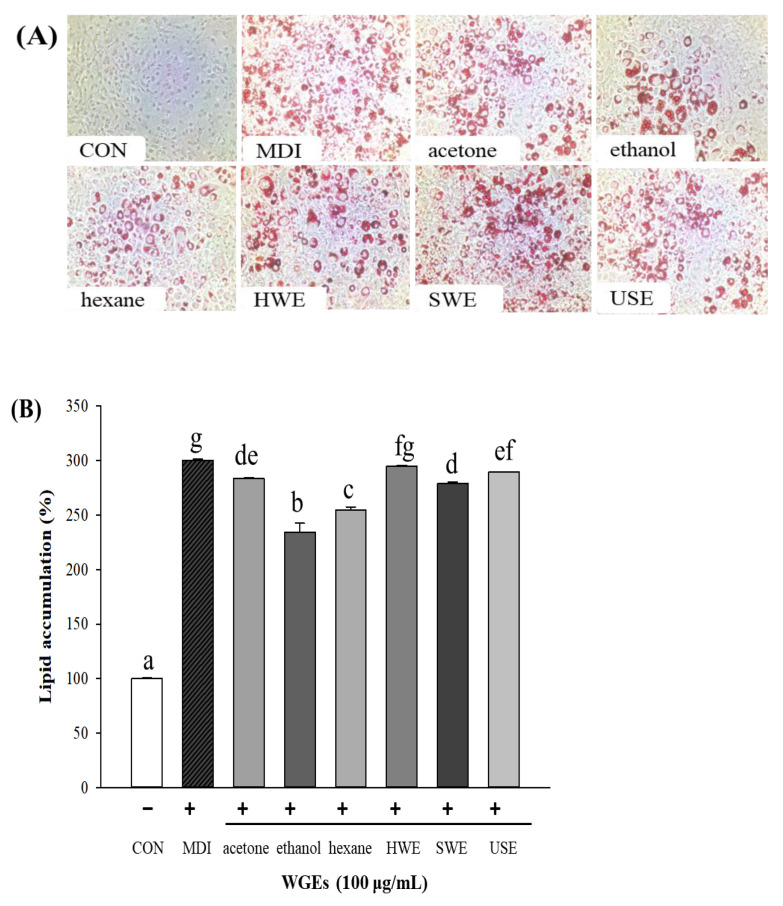
Effect of wheat germ extracts (WGEs) on cellular lipid accumulation in 3T3-L1 adipocytes. (**A**) The morphology of stained intracellular lipid droplets by Oil Red O staining at 400× magnification. (**B**) The content of cellular lipid accumulation stained by Oil Red O. Different letters indicate significant differences at *p* < 0.05, determined by Duncan’s multiple range test. CON, control; MDI (methylisobutylxanthine, dexamethasone, insulin; differentiated media); HWE, hot water extract; SWE, subcritical water extract; UWE, ultrasonic water extract.

**Figure 6 foods-10-01061-f006:**
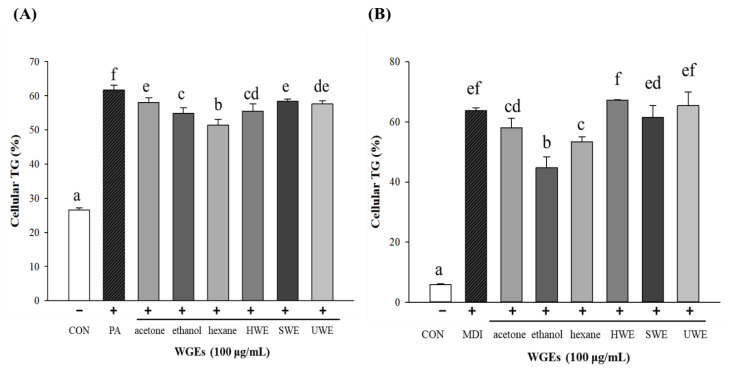
Effect of wheat germ extracts (WGEs) on cellular triglyceride (TG) accumulation. (**A**) The cellular TG accumulation in hepatocytes. (**B**) The cellular TG accumulation in differentiated adipocytes. TG concentration was normalized with protein content. Different letters indicate significant differences at *p* < 0.05 determined by Duncan’s multiple range test. CON, control; PA, palmitic acid; MDI (methylisobutylxanthine, dexamethasone, insulin; differentiated media); HWE, hot water extract; SWE, subcritical water extract; UWE, ultrasonic water extract.

**Figure 7 foods-10-01061-f007:**
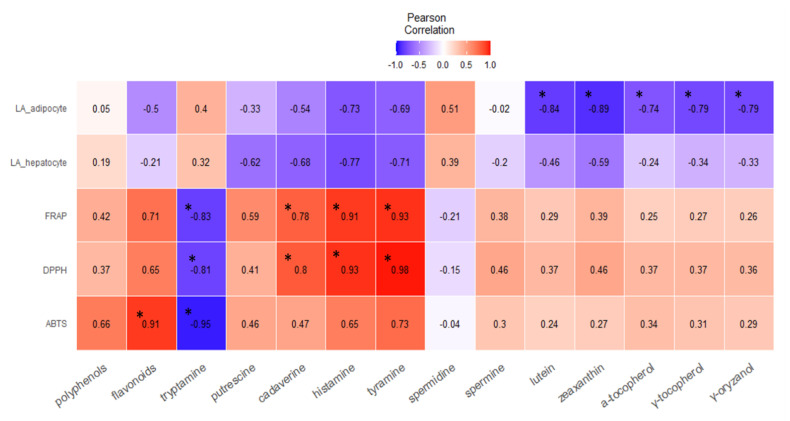
Heatmap showing Pearson’s correlation. Correlations between bioactive compounds and the antioxidant and antilipidemic activities of wheat germ extracts are shown. ABTS, DPPH, and FRAP show antioxidant activity, whereas the LA_hepatocyte and LA_adipocyte factors indicate an antilipidemic effect in those two cell types. The asterisk (*) shows significant differences (*p* < 0.05). LA_adipocyte, lipid accumulation in adipocytes; LA_hepatocyte, lipid accumulation in hepatocytes; FRAP, ferric reducing antioxidant power; DPPH, 2,2-diphenyl-1-picrylhydrazyl; ABTS, 2,2-azino-bis(3-ethylbenzothiazodine-6-sulfonic acid).

**Table 1 foods-10-01061-t001:** Nutrients content of wheat germ.

Water	Protein	Total Lipids	Ash	Carbohydrates	Fiber	Energy	
g/100 g						kcal	kJ
11.12	23.15	9.72	4.21	51.80	13.2	360	1506

**Table 2 foods-10-01061-t002:** Determination of total polyphenols and flavonoids in wheat germ extracts.

Extract	Total Polyphenols (mg GAE/g)	Total Flavonoids (mg CE/g)
Acetone	100.00 ± 0.55 ^b^	15.04 ± 0.36 ^d^
Ethanol	80.95± 0.55 ^d^	17.54 ± 0.35 ^b^
Hexane	84.19 ± 0.59 ^c^	16.29 ± 0.36 ^c^
HWE	79.49 ± 1.58 ^d^	12.24 ± 0.41 ^e^
SWE	128.62 ± 1.89 ^a^	21.29 ± 0.71 ^a^
UWE	67.60 ± 1.86 ^e^	9.38 ± 0.41 ^f^

The wheat germ extracts concentration used is 10 mg/mL. The total polyphenol content is expressed as milligrams of gallic acid equivalent (GAE) per g of extract. The total flavonoid content is expressed as milligrams of catechin equivalent (CE) per g of sample. Values are expressed as mean ± standard deviation (*n* = 3). Different letters in each column indicate statistically significant differences at *p* < 0.05 (Duncan’s multiple range test). HWE, hot water extract; SWE, subcritical water extract; UWE, ultrasonic water extract.

**Table 3 foods-10-01061-t003:** Determination of seven biogenic amines in wheat germ extracts by reversed-phase high-performance liquid chromatography.

	Concentration (mg/kg)						
Extract	Tryptamine	Putrescine	Cadaverine	Histamine	Tyramine	Spermidine	Spermine
Acetone	47.26 ± 1.39 ^b^	115.59 ± 0.80 ^e^	65.34 ± 1.75 ^a^	49.45 ± 0.22 ^b^	36.96 ± 1.21 ^b^	60.46 ± 0.64 ^b^	45.69 ± 0.56 ^b^
Ethanol	29.01 ± 0.14 ^d^	126.19 ± 0.97 ^d^	50.34 ± 1.27 ^c^	49.06 ± 0.19 ^b^	40.02 ± 0.24 ^a^	23.81 ± 0.45 ^c^	40.61 ± 0.21 ^c^
Hexane	23.70 ± 0.56 ^e^	194.16 ± 2.01 ^a^	54.82 ± 4.01 ^b^	50.16 ± 0.57 ^a^	35.67 ± 0.67 ^c^	17.71 ± 0.79 ^e^	38.56 ± 0.12 ^c^
HWE	39.14 ± 0.16 ^c^	152.75 ± 0.32 ^b^	ND	ND	16.45 ± 0.03 ^e^	234.91 ± 1.35 ^a^	62.87 ± 1.40 ^a^
SWE	9.22 ± 0.00 ^f^	137.27 ± 0.57 ^c^	34.05 ± 0.05 ^d^	24.76 ± 0.04 ^c^	17.57 ± 0.04 ^d^	19.54 ± 0.04 ^d^	13.41 ± 0.07 ^d^
UWE	229.35 ± 0.22 ^a^	99.25 ± 0.93 ^f^	16.47 ± 1.29 ^e^	ND	ND	11.59 ± 0.11 ^f^	7.73 ± 0.50 ^e^

The values are expressed as mean ± standard deviation (*n* = 3). Different letters in each column indicate statistically significant differences at *p* < 0.05 (Duncan’s multiple range test). ND, not detected; HWE, hot water extract; SWE, subcritical water extract; UWE, ultrasonic water extract.

**Table 4 foods-10-01061-t004:** R^2^ and LOD values of the validated HPLC method for biogenic amines in wheat germ extracts.

	Try	Put	Cad	His	Tyr	Spd	Spm
R^2^	0.9887	0.9919	0.9904	0.9816	0.9932	0.9978	0.9918
LOD	0.01	0.09	0.04	0.01	0.07	0.06	0.08

R^2^, linear correlation coefficient; LOD, limit of detection (mg/kg); Try, tryptamine; Put, putrescine; Cad, cadaverine; His, histamine; Tyr, tyramine; Spd, spermidine; Spm, spermine.

**Table 5 foods-10-01061-t005:** Determination of bioactive compounds in wheat germ extracts: carotenoids, tocopherols, and γ-oryzanol.

	Concentration (mg/kg)							
Extract	Lutein	Zeaxanthin	β-cry	α-car	β-car	α-toc	γ-toc	γ-oryzanol
Acetone	5.83 ± 0.07 ^d^	10.41 ± 0.41 ^c^	ND	ND	ND	ND	ND	ND
Ethanol	79.67 ± 0.38 ^a^	69.86 ± 1.70 ^a^	ND	ND	ND	312.17 ± 4.10 ^a^	228.37 ± 3.12 ^a^	5928.89 ± 23.32 ^a^
Hexane	24.5 ± 0.69 ^b^	31.60 ± 1.28 ^b^	ND	ND	ND	5.92 ± 5.08 ^b^	34.00 ± 1.21 ^b^	858.56 ± 9.21 ^b^
HWE	ND	ND	ND	ND	ND	ND	ND	ND
SWE	ND	ND	ND	ND	ND	ND	ND	ND
UWE	14.94 ± 0.33 ^c^	13.02 ± 1.45 ^c^	ND	ND	ND	ND	22.78 ± 0.30^c^	486.49 ± 7.43 ^c^

The values are expressed as mean ± standard deviation (*n* = 3). Different letters in each column indicate statistically significant differences at *p* < 0.05 (Duncan’s multiple range test). ND, not detected; HWE, hot water extract; SWE, subcritical water extract; UWE, ultrasonic water extract; β-cry, β-cryptoxanthin; α-car, α-carotene; β-car, β-carotene; α-toc, α-tocopherol; γ-toc, γ-tocopherol.

**Table 6 foods-10-01061-t006:** R^2^ and limit of detection values of the validated HPLC method for bioactive compounds: carotenoids, tocopherols, and γ-oryzanol in wheat germ extracts.

	Lut	Zea	β-cry	α-car	β-car	α-toc	γ-toc	γ-ory
R^2^	0.9992	0.9977	0.9993	0.9932	0.9982	0.9945	0.9936	0.9852
LOD	0.03	0.05	0.07	0.01	0.01	0.01	0.02	0.02

R^2^, linear correlation coefficient; LOD, limit of detection (mg/kg); Lut, lutein; Zea, zeaxanthin; β-cry, β-cryptoxanthin; α-car, α-carotene; β-car, β-carotene; α-toc, α-tocopherol; γ-toc, γ-tocopherol; γ-ory, γ-oryzanol.

**Table 7 foods-10-01061-t007:** Determination of antioxidant activities from wheat germ extracts.

Extract	ABTS (%)	DPPH (%)	FRAP Value
Acetone	74.23 ± 0.38 ^d^	66.97 ± 0.35 ^a^	0.80 ± 0.00 ^b^
Ethanol	83.71 ± 0.38 ^b^	64.53 ± 0.77 ^b^	0.78 ± 0.01 ^b^
Hexane	78.14 ± 0.63 ^c^	63.92 ± 0.47 ^b^	0.95 ± 0.00 ^a^
HWE	68.88 ± 1.14 ^e^	36.02 ± 1.04 ^d^	0.50 ± 0.03 ^d^
SWE	87.54 ± 0.25 ^a^	46.83 ± 1.25 ^c^	0.69 ± 0.03 ^c^
UWE	40.49 ± 1.56 ^f^	10.54 ± 1.61 ^e^	0.20 ± 0.02 ^e^

The wheat germ extracts concentration used is 10 mg/mL. Different letters in each column indicate statistically significant differences at *p* < 0.05 (Duncan’s multiple range test). ABTS, 2,2-azino-bis(3-ethylbenzothiazodine-6-sulfonic acid); DPPH, 2,2-diphenyl-1-picrylhydrazyl; FRAP, ferric reducing antioxidant power; HWE, hot water extract; SWE, subcritical water extract; UWE, ultrasonic water extract.

## Data Availability

The data presented in this study are available on request from the corresponding author.

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
