# Peer review of "Antioxidant Effect of Wheat Germ Extracts and Their Antilipidemic Effect in Palmitic Acid-Induced Steatosis in HepG2 and 3T3-L1 Cells"

_foods, 2021, doi:10.3390/foods10051061_

Round 1

Reviewer 1 Report

The manuscript prepared by Liaqat et al. was exploring wheat germ extracts (WGE) and the bioactive compounds, as well as the antioxidant effects of WGEs and the antiadipogenic effects using two cell lines. The manuscript is relatively comprehensive and experiment design is reasonable. The reviewer suggests the authors to address the points before re-submitting to our journal.

General comments:

  1. The English language should be improved through the manuscript since some parts were not easy to read.
  2. In the introduction, please revise it and emphasize the novelty. For example, the importance of biogenic compounds in WGEs. Do you have any potential application and/or suggestion in food industry?
  3. In the materials, Where were the wheat germs samples from? How to store the samples?
  4. What did you mean for '2.6 Linearity and Lower Limit of Detection (LOD)'?
  5. In tables, please note what were 'HWE', 'UWE', and 'SWE'.
  6. In line 152, please specific the 'sample extract (prepared above)'.
  7. In Figure 2, Please specific which sample you use for '(B) Representative chromatogram of one of the samples.'

Reviewer 2 Report

The MS deals with the partial chemical characterization of some wheat germ extracts and the analysis of some biological analysis. The study is interesting, elucidating some important feature of this byproducts. However, some issues should be addressed.

The Authors should state in the Methods how they acquired the wheat germ, and possibly some of its characteristics (for instance the moisture %).

Besides, in the Abstract they state that WG is a by-product. But this aspect is not reported in the Introduction: in my opinion this should be on the contrary well explained in the Introduction to add value to the whole experimental plan.

Line 115: all the lyophilized WGEs were solubilized in water for the further analysis? No precipitation occured also with organic solvent extracts?

Tables 2 and 3: the Authors determined SD using only 2 repetitions: is it correct? It is quite strange to use less than n = 3…

Table 4: the Authors decided to express the antioxidant effect as % inhibition , and not using a standard. However, in Literature TROLOX as the standard is widely used in such assays, allowing to compare different studies and different matrix. 

Why did the Authors not choose this option? Do they have these experiments expressed as Trolox equivalents per mg of extract?

Lines 457-480: this incipit should be greatly summarized. In general, all the discussion focuses on too broad topic (just for instance, see also lines 490-520), without really discussing the results of this study, and eventually comparing them with Literature. I suggest to greatly rewrite this part.

MINOR POINTS:

Line 64: please, insert a space after the word “benefits”

Figure 1: all the formula should have the same style (some fonts for instance seem bigger, and also not all chemical bonds have the same thickness)

Round 2

Reviewer 1 Report

The authors have corrected the points suggested by the reviewer.

Author Response

Thank you for your review.

Reviewer 2 Report

The Authors effectively responded to the criticisms in R1, improving the quality of the MS.

Author Response

Thank you for your review.